# Personal Care Products Are Only One of Many Exposure Routes of Natural Toxic Substances to Humans and the Environment

**Thomas D. Bucheli [1],\*, Bjarne W. Strobel [2] and Hans Chr. Bruun Hansen [2]** 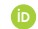

[1]   Agroscope, Environmental Analytics, Reckenholzstrasse 191, 8046 Zürich, Switzerland
[2]   Department of Plant and Environmental Sciences (PLEN), University of Copenhagen, Thorvaldsensvej 40, 1871 Frederiksberg, Copenhagen, Denmark; bjwe@plen.ku.dk (B.W.S.); haha@plen.ku.dk (H.C.B.H.)
\*   Correspondence: thomas.bucheli@agroscope.admin.ch; Tel.: +41-58-468-7342

**Abstract:** The special issue "A Critical View on Natural Substances in Personal Care Products" is dedicated to addressing the multidisciplinary special challenges of natural ingredients in personal care products (PCP) and addresses also environmental exposure. In this perspective article, we argue that environmental exposure is probably not so much dominated by PCP use, but in many cases by direct emission from natural or anthropogenically managed vegetation, including agriculture. In support of this hypothesis, we provide examples of environmental fate and behaviour studies for compound classes that are either listed in the International Nomenclature of Cosmetics Ingredients (INCI) or have been discussed in a wider context of PCP applications and have been classified as potentially harmful to humans and the environment. Specifically, these include estrogenic isoflavones, the carcinogenic ptaquiloside and pyrrolizidine alkaloids, saponins, terpenes and terpenoids, such as artemisinin, and mycotoxins. Research gaps and challenges in the domains of human and environmental exposure assessment of natural products common to our currently rather separated research communities are highlighted.

**Keywords:** phytotoxins; mycotoxins; micropollutants; drinking water

---

## 1. Introduction

Human and environmental risks of xenobiotic toxic substances in personal care products (PCPs) have been acknowledged and studied for quite a while [1–6]. In addition to these anthropogenic chemicals, Klaschka [7] identified over one thousand natural substances—mainly of herbal origin—that appear in the International Nomenclature of Cosmetics Ingredients (INCI), and out of which 27% are classified as hazardous to human and environmental health. Already in 2006, the Committee of Experts on Cosmetic Products had listed 24 potentially harmful plant components in cosmetics [8]. It has also been pointed out that the classification, regulation, and risk assessment of toxic natural products used in cosmetics is weak, incoherent, and inconsistent [7], and hence there may be situations with no or poor human and/or environmental protection. Even the European Regulation on the Registration, Evaluation, Authorisation and Restriction of Chemicals (REACH) does in most cases not include natural substances, although some of them are known to be toxic. There is no doubt that some of the ingredients used in cosmetics in large amounts, such as soaps, shampoo, creams, and lotions, may contain natural compounds that pose a risk both to human health and to the aquatic environment. Examples are cyanogenic glucosides, different toxic alkaloids (such as strychnine, atropine, and aconitine), phytoestrogens, such as sterols and isoflavones, terpenes and terpenoids (as in many oils and fats), quinones, and peroxides. Based on the fact that almost 200 compounds have been categorized as harmful to very toxic to aquatic life, some of which with long- or very-long-lasting

effects, and that the use of a natural cosmetic product as a PCP implies wide dispersive use and continuous discharge via local discharges or sewer systems into the aquatic environment, the release of natural compounds from PCPs to the environment needs to be assessed [7,9].

Consequently, the special issue "A Critical View on Natural Substances in Personal Care Products "addresses among other challenges the "direct and indirect human and environmental exposure" to natural compounds in PCPs. However, natural compounds are ubiquitous in the environment as most living organisms produce at least one natural compound that is bioactive and which may function to protect the organism against predators and pathogens or as a chemical warfare agent in competition with other species. Thus, as the natural bioactive (and many toxic) compounds produced by plants, micro-organisms, and animals are expected to be widely present, although very seldom monitored, there are many potential exposure routes for natural compounds, of which PCPs are (only) one. In the following, we outline these different sources of toxic compounds, and provide some illustrative examples of natural compounds that are present in PCPs and have been investigated in the environment.

## 2. Routes of Human and Environmental Exposures to Natural (Toxic) Substances

Human exposure to natural toxic substances can occur via direct physical (dermal or oral) contact with toxic plants and their active ingredients (Figure 1, black arrow) via PCPs, food, and drinking water (Figure 1, various arrows). The importance of natural toxic substance intake by food in particular is well-known [10]. The environmental risk of natural toxic substances by PCP consumption and emission via sewers may be relevant in some cases (Figure 1, blue arrows), but is probably not the predominant exposure pathway. Elimination rates in wastewater treatment plants (WWTP) are high for many micropollutants [11] and are further increased by additional cleaning steps, such as activated carbon or ozonation [12]. Direct emissions of (natural) PCP from the anthroposphere into the aqueous environment, e.g., from industry, households, and small settlements not connected to WWTP, during flooding events, and by bathing activities, may occur [13] but probably lead to relatively low exposures.

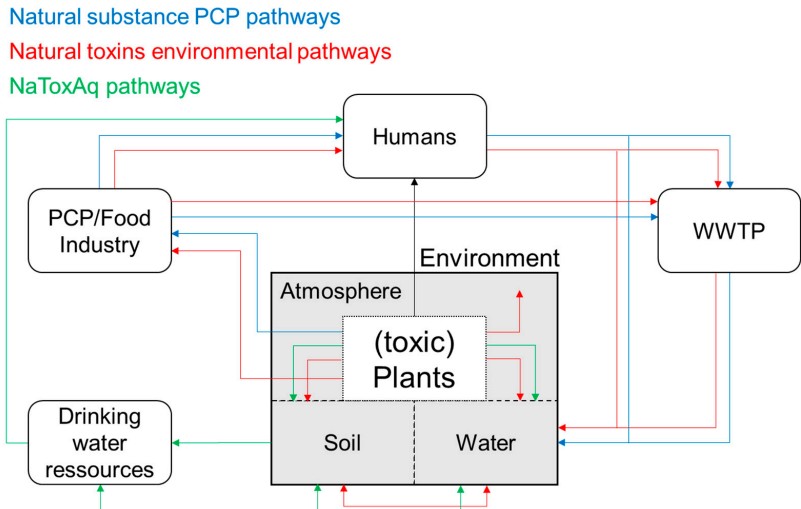

**Figure 1.** Major pathways and lines of investigation of (toxic) natural compounds from plants at the interface of the anthroposphere (conceptually presented as round-corner boxes) and the environment (grey rectangles). Blue arrows indicate substance flows of natural substances used in Personal Care Products (PCPs, proposed by Klaschka [9], and the focus of this special review covered by other references), red arrows focus on fate and behaviour of natural toxins in the environment [14], and green arrows represent domains covered by the ongoing European Training Network Project "Natural Toxins and Drinking Water Quality—From Source to Tap (NaToxAq)" [15] (for details, see text). WWTP: wastewater treatment plant.

With regard to environmental exposure, we consider direct input from vegetation and agricultural areas as the dominant input pathway (Figure 1, red arrows). Phytotoxins in particular have been suggested as potential aquatic micropollutants that should be assessed systematically [14]. The human risks of natural (toxic) substances via drinking water are currently being addressed in a European Training Network (ETN) Project called "Natural Toxins and Drinking Water Quality—From Source to Tap (NaToxAq)", which is funded by the European Union's Horizon 2020 research and innovation programme [15].

## 3. Examples of (Direct) Environmental Exposure to Natural (Toxic) Substances

### 3.1. Phytoestrogens

Phytoestrogenes, such as coumestrol and isoflavones, and in particular genistein (Figure 2) and daidzein from soy, are often considered beneficial for human health, e.g., by reducing risks for breast cancer [16], by alleviating menopausal symptoms, or by helping to prevent cardiovascular diseases and osteoporosis [17]. However, some authors draw a more ambivalent, and not yet final, picture [18,19]. After all, these compounds are also known to cause breeding problems in sheep [20] or infertility in captive cheetahs [21]. Nevertheless, isoflavones are widely used in healthcare products, e.g., as active ingredients in skin care cosmetics [22], due to their antioxidant and antipromotional [23], rejuvenating [24], photoprotective [25], and anti-inflammatory [26] properties.

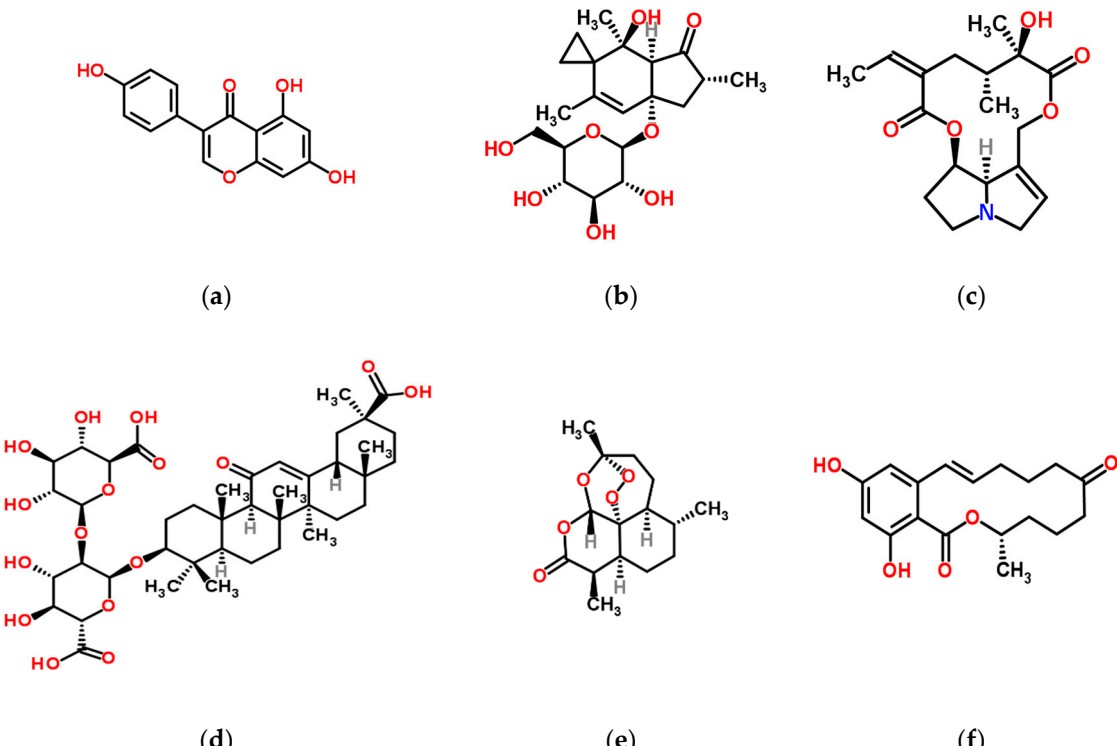

**Figure 2.** Chemical structures of selected natural toxic compounds that are present in personal care products and potentially relevant for the environment (all structures from http://www.chemspider.com): (**a**) genistein (an isoflavone), (**b**) ptaquiloside (a norsesquiterpene glycoside), (**c**) senecionine (a pyrrolizidine alkaloid), (**d**) glycyrrhizin (a saponine), (**e**) artemisinin (a terpenoid), and (**f**) zearalenone (a resorcyclic acidic lactone).

As a consequence of their regular consumption via food and healthcare products, they are excreted and emitted via WWTP to the aqueous environment [27,28]. An alternative, and potentially even more important, source of isoflavones to the environment are agricultural areas cultivated with forage and

grain legumes, as such plants can produce up to 220 kg of isoflavones per hectare and year [29]. Indeed, formononetin, the dominating isoflavone compound in red clover, was detected most frequently in Swiss [28] and U.S. surface waters [30].

### 3.2. Ptaquiloside

Ptaquiloside—a carcinogenic norsesquiterpene glycoside (Figure 2)—is produced by Bracken (*Pteridium aquilinum*), one of the most abundant plants on Earth, which exhibits invasive properties outcompeting other vegetation [31,32]. Bracken, which is registered as a plant extract in the INCI database (CAS 90105-98-9), has been patented for use in cosmetics, e.g., as a skin conditioner, moisturising milk, facial mask, or cleansing cream for skin scrubbing, including the use of bracken spores (e.g., [33–37]). It is also used in foods and medicine, mainly in Asia, and is a known contaminant in milk produced by cows, sheep, and goats browsing on bracken [38]. There is ample evidence to show its presence in surface [39] and groundwater [40], and thus ptaquiloside may eventually contaminate drinking water (Figure 1). Although the suspected carcinogenicity of ptaquiloside has been known for a long time and there are epidemiological studies that suggest it to cause esophageal and stomach cancer in humans [41,42], there is little or no monitoring, regulation, and control of ptaquiloside in strong contrast to the regulation of, e.g., pesticides in drinking water, even though they may not be as toxic as ptaquiloside. Ptaquiloside is not registered in REACH. Ptaquiloside may constitute up to 5% of bracken dry weight for leaves, resulting in loads of up to 100 kg per hectare and year, which is orders of magnitude higher than the loads of, e.g., pesticides. Ptaquiloside is readily washed out of the plant by precipitation, and because of its polar nature (Figure 2), it is poorly retained in soils and thus passes on to water reservoirs. In fact, many natural compounds occur as glucosides or conjugated to polar groups [43], which makes the compounds highly mobile and easily spread in the environment, thus also increasing human exposure via water.

### 3.3. Pyrrolizidine Alkaloids

Pyrrolizidine alkaloids, such as senecionine (Figure 2) and senkirkine, are common constituents in many plants, e.g., *Senecio jacobaea*, *Echium vulgara*, and *Tussilago farfara*, where they may function as insecticides [44]. Pyrrolizidine compounds are not in the INCI list, but extracts of plants such as *Senecio vulgaris*, *Symphytum officinalis*, and *Tussilago farfara*, which are known to produce pyrrolizidine alkaloids, are used in cosmetics [8]. Pyrrolizidine alkaloids are known to be both mutagenic and carcinogenic. Nectar and pollen contain pyrrolizidine alkaloids that are transferred to honey as a contamination by this natural toxic compound [45]. Concentrations of pyrrolizidine alkaloids in honey are found up to 95 $\mu$g g$^{-1}$, and for pollen totals to 35,000 $\mu$g g$^{-1}$ [46]. Pollen and nectar addition to PCPs embrace the healthy nature of the bees collecting all the "good" natural substances in flowers, and for pollen products, the pyrrolizidine alkaloids come along as unintended contamination. To the best of our knowledge, pyrrolizidine alkaloids have not yet been studied in the aqueous environment, but based on their chemical–physical properties, their appearance seems plausible [14].

### 3.4. Saponins

Saponins are natural surfactants comprised of a triterpenoid or steroid aglycone conjugated to one or several sugar units (Figure 2) [47,48]. Saponins are present in substantial amounts (up to 30% in liquorice root) in a large diversity of plants, such as the soap bark tree (*Quillaja saponaria*), common soapwort (*Saponaria officinalis*), *Yucca sp.*, liquorice (*Glycyrrhiza glabra*), ginseng (e.g., *Panax ginseng*), and many crops, such as quinoa, alfalfa, beans, and peas (Table 1). Saponins are popular foaming, emulsifying, and cleansing ingredients in shampoos, shower gels, and bath foams, but also as moisturising ingredients in creams and lotions. Saponins are registered in the INCI list with different CAS entries, and there are several hundred registered patents (Derwent database) on the use of saponins and saponin extracts in cosmetics. Saponins act as defense compounds in leaves [49]. The antimicrobial effects of saponins are well-documented (e.g., [50]). Saponins also find use in

medicine and as biopesticides (e.g., [51,52]), and even as a solubilizer for the remediation of non-ionic contaminants in soils [53]. The toxic effects of saponins are related to their surfactant functions, which may cause a disruption of cell membranes through interaction with cell membrane lipid components, such as cholesterol, and cell lysis may result. Saponins are highly water soluble, and because of their surfactant properties they are solubilizers of lipophilic non-ionic compounds, such as waxes and fats. Recent studies have shown that saponins are toxic to a wide variety of organisms in aquatic systems, that different organisms exhibit widely different sensitivities to toxication, and that the toxicity is strongly depending on the specific molecular structure, making "read-across" between saponins difficult. Apparently, saponins are rather stable compounds at neutral to slightly acid conditions, such as those found in many lake waters [54]. For cosmetics, the main concern is considered to be the direct exposure of the skin to saponins during use, while saponins discharged to municipal wastewater are expected to degrade during wastewater treatment.

## 3.5. Terpenes and Terpenoids

Terpenes and terpenoids are common in needles of pine, cedar, spruce, and most coniferous trees, adding a distinct odour to coniferous forests during summer when the temperature favours the volatile compounds to fill the air [55,56]. Western red cedar and wormwood are known to produce thujone in addition to several more terpenes [49,50]. The essential oils extracted from, e.g., sage, lavender, and pine add a pleasant flavour of freshness and clean air aspiration to shampoo and lotions, and household cleaning products [57] contain natural terpenes and terpenoids. Some are potentially allergenic and end up in freshwaters after showers and from washing clothes that have wiped off the lotion from the skin. Terpenes are used as environmentally friendly repellents to insects [58]. The tropolone compound thujapline, also known as hinokitiol, adds anti-fungal and anti-bacterial properties to the western red cedar hard wood [59,60] and has been used in cosmetics for a long time [61]. Sweet wormwood (*Artemisia annua*) produces the active compound artemisinin (Figure 2), which is a sesquiterpene with a peroxide group. The compound is known from traditional Chinese medicine in China and is grown in several continents, and its impact on the soil and water environment has been studied in Europe and the U.S. [62–64].

## 3.6. Mycotoxins

Natural toxic compounds are not only produced by plants, but also by bacteria, algae, animals, or fungi [65]. Mycotoxins, such as aflatoxins, trichothecenes, fumonisins, zearalenone (Figure 2), ochratoxins, and ergot alkaloids, produced by several species of the fungal genera *Aspergillus*, *Fusarium*, *Penicillium*, and *Claviceps*, are of concern for human and husbandry animal health. Consequently, many of them they are regulated by authorities with maximum levels (e.g., [66]), indicative levels (e.g., [67]), guidance values (e.g., [68]), or tolerable daily intake levels (e.g., [69]), and are regularly monitored in food and feed products. Traces of these compounds are excreted after consumption, and can enter surface water after only a partial elimination in WWTP [27,70,71]. Conversely, field-produced mycotoxins are directly emitted into surface waters via runoff and drainage from *Fusarium*-infected agricultural fields [72–75]. In such cases, the prevention of fungal infections will directly lead to a decrease in environmental mycotoxin exposure.

In summary, these examples illustrate that while many natural bioactive compounds/toxins are ingredients in PCPs, they are also and already (naturally) present in terrestrial or aquatic compartments and ecosystems. Hence, PCPs may add to the aqueous environment's exposure to natural (toxic) substances, but probably with a rather low share. Much higher loads come from other (natural or anthropogenically managed) sources. With regard to human health, the more important concern for PCPs is the direct exposure to natural toxins, first of all through the skin.

## 4. Common Ground and Future Research Needs

Research communities that investigate natural (toxic) substances in PCPs and the environment currently seem to be rather disparate, although they share a common interest in the same compounds. Table 1 lists examples of natural substances that have been addressed by both communities. Information on the chemical composition in plants or plant extracts, chemical–physical properties, (eco-)toxicity, and reactivity of natural (toxic) substances would be valuable for researchers active in both fields. Certified chemical standards, labelled standards, and reference samples are often missing, and analytical methods for the quantification of individual analytes at adequate concentration ranges are yet to be established.

**Table 1.** Examples of (toxic) natural substances mentioned in cosmetics and studied in the environment.

| Natural (Toxic) Substance(s) | Examples of Producing Species (Plants or Fungi) | Discussed in the Context of Personal Care Products | Studied in the Environment |
|---|---|---|---|
| Coumestrol | *Medicago sativa* | [8,76] | [77] |
| Isoflavones | *Trifolium pratense* | [8,9] | [28,29] |
| Limonene, pinenes | *Citrus spp.* | [9,78] | [55,79–81] |
| Linalool | *Lavandula sp.* | [82,83] | [55] |
| Peroxides | *Artemisia annua* | [9] | [62–64] |
| Ptaquiloside | *Pteridium aquilinum* | [33–37] | [39,40] |
| Pyrrolizidine alkaloids | *Tussilago farfara* | [84] | Not yet, occurrence possible [14] |
| Saponins | *Yucca schidigera* | [85] | [49,54] |
| Thujone | *Thuja plicata* *Artemisia absinthium* | [7,8] | [81,86] |
| Thujaplicine | *Thuja plicata* | [61,87] | [59,88] |
| Zearalenone | *Fusarium graminearum* | [9] | [73,75] |

With regard to environmental exposure to natural (toxic substances), information about the spatial distribution of toxic plants, and the corresponding stocks of natural toxins in relation to surface and groundwater catchments, would be needed. In many cases, we have yet to establish robust and representative sampling and sample preservation methods [89]. Tools for the prioritization of the (eco-)toxicologically most relevant among the myriad of natural substances are needed to efficiently use limited resources. A corresponding sequence of modules was proposed elsewhere [14], and is currently under development within the NaToxAq network [15]. There is no current overview of which natural toxins may be present in environmental matrices, and in fact many natural toxins, including their conjugates and metabolites, are yet to be identified. In NaToxAq, both *in silico* approaches as well as non-targeted, targeted, and effect-directed analyses are used to identify natural toxins in drinking water resources, and then to conceptualise, quantify, and eventually model the flow and fate of natural toxins from their sources to drinking water recipients. The fundamental insight generated is expected also to be important for an assessment of the fate of natural toxins originating from PCPs. As such, NaToxAq provides a platform for interaction and joining forces in our quest to elucidate human and environmental exposure to natural toxic substances.

**Acknowledgments:** Funding of the Swiss National Science Foundation (Project "PHYtotoxins: aquatic miCROPOLLutants of concern?" (PHYCROPOLL); Grant No. 200021_162513/1) and of the Marie Curie Innovative Training Network "Natural Toxins and Drinking Water Quality—From Source to Tap (NaToxAq)" (Grant No. 722493) by the European Commission is gratefully acknowledged. We are grateful to the members of this consortium for many fruitful discussions. Finally, we thank the Special Editor Ursula Klaschka for her initiative, for crossing scientific borders and for bringing natural substances in PCPs to our attention.

**Author Contributions:** The authors jointly elaborated the fundamental line of argumentation of this perspective paper. Thomas D. Bucheli acted as lead author, drafted the figures and tables, wrote individual compound class cases, and was the leading author for the introduction and the perspectives part. Bjarne W. Strobel contributed individual compound class cases. Hans Chr. Bruun Hansen wrote individual compound class cases and contributed to the introduction and the perspectives part of the paper.

**Conflicts of Interest:** The authors declare no conflict of interest.

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
