# Peer review of "Personal Care Products Are Only One of Many Exposure Routes of Natural Toxic Substances to Humans and the Environment"

_cosmetics, doi:10.3390/cosmetics5010010_

Round 1

Reviewer 1 Report

see attached file

Author Response

See also attached file...

Reviewer 1:

The points made by the manuscript and the overall scope are easy to appreciate. Overall, the manuscript is brief and to-the-point. It could be argued that this is exactly what's needed for a Cosmetics perspective.

Many thanks for this overall very positive feedback!

Having said this, I would like to bring up issues not yet raised. Perhaps they could be briefly addressed here, or instead the authors could include them in later manuscripts where being brief is less of an issue:

- The manuscript does not cover natural products in the gas phase. Volatile exposures are important in some situations, e.g. "sick buildings" arising from volatiles from mildew and mold.

The reviewer is right, but we consider this somewhat outside the scope of this manuscript, which focuses on (anthropogenially managed) vegetation and agriculture (see lines 14-15, 69-70, 77-79), and compounds that have been mentioned in the International Nomenclature of Cosmetics Ingredients (INCI) or have been discussed in a wider context of PCP applications (lines 17-18). Nevertheless, gaseous emissions (of, e.g., terpenes and terpenoids) are indicated in Figure 1 and mentioned specifically in lines 163-165.

- There is growing concern about natural product exposures from materials, e.g. shipping boxes and pallets. I'm not sure if anyone has investigated them yet, but sphagnums and other fillers might be an issue, as well as molds, mildews, etc. transported by soils on potted plants.

These are interesting aspects, but again, we have difficulties to see how they could be integrated here.

- There is growing recognition that we need to explore beyond parent compounds. Disinfection reactions (and other chemical reactions) yield new chemicals, some of which are considerably more toxic than their parent compounds. In some instances benign compounds are chemically converted into toxic compounds. Nitrogen-containing disinfection by-product precursors, leading to the formation of nitrosamines is an example of this.

The reviewer is absolutely right. Metabolites of anthropogenic and natural compounds are understudied and should receive more (scientific) attention. This is mentioned as a future research need in line 214.

- In some situations, lessening natural product exposures will require management of organisms rather than the natural products themselves. Mildews and molds in buildings are obvious examples. Fungal toxin production in agricultural residues might be another.

Again, we fully agree with the reviewer. To address the reviewer’s point, we have added in line 187-188: “In such cases, prevention of fungal infections will directly lead to a decrease in environmental mycotoxin exposure.”

Details:

line 101 Owing to its prominent place in the text (line 98) it might be a good idea to include the structure of formonometin in Figure 2.

While we agree with the reviewer about the overall importance of formononetin, we prefer to present the chemical structure of genistein in Figure 2, another example of isoflavones, because it is probably even better known to most readers. Generally, figure 2 contains one example of each of the cases presented in the chapters 3.1 to 3.6, and adding a second compound would give chapter 3.1 too much weight.

line 140 replace "composed" with "comprised"

Done as requested.

line 141 Physicochemical properties of natural products are important for their environmental behavior and toxicology. The authors might want to highlight their importance at some point. Here, for example, processes adding or substracting sugar units can dramatically alter solubility, volatility, and bioavailability.

We fully agree with the reviewer about the importance of the chemical’s physical-chemical properties, both in this particular case, and in general. Therefore, chemical-physical properties are mentioned in line 198, as an example of valuable information for researchers in the field.

Line 154 replace "vaxes" with "waxes"

Many thanks for bringing this typing error to our attention. Corrected as requested.

Reviewer 2 Report

The above noted manuscript has been submitted for possible publication in the Cosmetics journal. The manuscript refers to the occurrence of natural substances present as ingredients of some cosmetics and their occurrence in the environment as two sources of exposure of humans to these substance and their possible toxic effects. This is a very interesting issue and of high importance. In fact, many of these substances can attain high concentration in the environment not only by consumption of these products but also has natural occurring compounds of many plants and are able to cause adverse effects to exposed organisms. Authors selected most common used compounds in cosmetics listed in the International Nomenclature of Cosmetics Ingredients (INCI). Nevertheless authors should clarify if there is studies even in other species that demonstrates toxicity of the presence/occurrence of these natural substances and if there is possible risks to humans.  Thus, I suggest the acceptance of the manuscript for publication in the Cosmetics after some minor revisions.

Author Response

see also attached file...

Reviewer 2:

The above noted manuscript has been submitted for possible publication in the Cosmetics journal. The manuscript refers to the occurrence of natural substances present as ingredients of some cosmetics and their occurrence in the environment as two sources of exposure of humans to these substance and their possible toxic effects. This is a very interesting issue and of high importance. In fact, many of these substances can attain high concentration in the environment not only by consumption of these products but also has natural occurring compounds of many plants and are able to cause adverse effects to exposed organisms. Authors selected most common used compounds in cosmetics listed in the International Nomenclature of Cosmetics Ingredients (INCI). Nevertheless authors should clarify if there is studies even in other species that demonstrates toxicity of the presence/occurrence of these natural substances and if there is possible risks to humans.  Thus, I suggest the acceptance of the manuscript for publication in the Cosmetics after some minor revisions.

We acknowledge the positive general feedback of reviewer 2, but have difficulties to understand his/her request to “clarify if there is studies even in other species that demonstrates toxicity of the presence/occurrence of these natural substances and if there is possible risks to humans”.

Certainly, there are many further (e.g., plant, fungal, or bacterial) species that produce natural (toxic) compounds, which are not covered by the INCI. They are, however, not within the scope of this manuscript, and have been reviewed or investigated elsewhere (e.g. ref 14 & 15 in our paper).

Regarding the (eco-)toxicity of natural substances and their risk to humans, these aspects are briefly addressed in the chapters 3.1 to 3.6, and in the papers cited therein. Specifically, ecotoxicity of phytoestrogens (chapter 3.1) are mentioned in lines 90-91, toxicity of ptaquiloside (chapter 3.2) in lines 115-117, toxicity of pyrrolizidine alkaloids (chapter 3.3) in lines 130-131, ecotoxicity of saponins (chapter 3.4), in lines 148-153, and 155-158, ecotoxicity of terpenes (chapter 3.5) in lines 170-172, and toxicity of mycotoxins (chapter 3.6) in line 181.

Overall, we therefore consider the request by reviewer 2 to be generally addressed in the article, but are open to further (more specific) amendments if they are still considered useful.
